# Monocular Camera Viewpoint-Invariant Vehicular Traffic Segmentation and Classification Utilizing Small Datasets

**DOI:** 10.3390/s22218121

**Published:** 2022-10-24

**Authors:** Amr Yousef, Jeff Flora, Khan Iftekharuddin

**Affiliations:** 1Engineering Mathematics Department, Alexandria University, Lotfy El-Sied st. off Gamal Abd El-Naser, Alexandria 11432, Egypt; 2Electrical Engineering Department, University of Business and Technology, Ar Rawdah, Jeddah 23435, Saudi Arabia; 3Electrical Engineering Department, Old Dominion University, 5115 Terminal Blvd, Norfolk, VA 23529, USA

**Keywords:** low-rank, matrix decomposition, vehicle segmentation and classification, HOG, Kalman filter, SVM, deep learning architectures

## Abstract

The work presented here develops a computer vision framework that is view angle independent for vehicle segmentation and classification from roadway traffic systems installed by the Virginia Department of Transportation (VDOT). An automated technique for extracting a region of interest is discussed to speed up the processing. The VDOT traffic videos are analyzed for vehicle segmentation using an improved robust low-rank matrix decomposition technique. It presents a new and effective thresholding method that improves segmentation accuracy and simultaneously speeds up the segmentation processing. Size and shape physical descriptors from morphological properties and textural features from the Histogram of Oriented Gradients (HOG) are extracted from the segmented traffic. Furthermore, a multi-class support vector machine classifier is employed to categorize different traffic vehicle types, including passenger cars, passenger trucks, motorcycles, buses, and small and large utility trucks. It handles multiple vehicle detections through an iterative k-means clustering over-segmentation process. The proposed algorithm reduced the processed data by an average of 40%. Compared to recent techniques, it showed an average improvement of 15% in segmentation accuracy, and it is 55% faster than the compared segmentation techniques on average. Moreover, a comparative analysis of 23 different deep learning architectures is presented. The resulting algorithm outperformed the compared deep learning algorithms for the quality of vehicle classification accuracy. Furthermore, the timing analysis showed that it could operate in real-time scenarios.

## 1. Introduction

Computer vision methods may offer more practical perceptual, contextual, and situational awareness to a highway intelligent transportation system (ITS) than any other types of sensors [1]. Utilizing vision sensors in the development of the ITS focuses on challenges that are generally difficult to solve using other sensor approaches [2]. For example, vehicles come in various sizes, shapes, and colors within the same class. Additionally, the projected shape of the vehicle differs due to pose variations concerning the camera sensor. All the previously mentioned points make the problem of vehicle detection and classification very challenging. Another complexity in ITS arises due to highway traffic outdoor scenes with varying visibility and lighting conditions. Furthermore, limited computing power may reduce the efficiency of an integrated system to perform in real time. Parallel computing techniques have decreased this limitation drastically. Finally, accounting for the unpredictable movement of vehicles through a highway traffic scene may be difficult. Off-road cameras are currently used by most state departments of transportation to monitor traffic conditions and to identify situations that disrupt traffic flow, such as road debris, stopped vehicles, or accidents. Computer vision algorithms allow these tasks to be performed automatically and efficiently [3].

Tremendous efforts to tackle different challenging aspects have been investigated deeply in the literature. While there are several other non-vision sensor-based approaches, such as the one presented in [4], the authors in this work focused on computer vision and machine-learning-related techniques utilizing monocular optical sensors.

The authors in [5] proposed a pipeline for detecting vehicles from monocular videos based on transfer learning. They weakly calibrated the camera using a 3 × 3 transformation from the image domain to the real-world domain to estimate the vehicle length and speed. Additionally, the authors adopted three popular object detection approaches, namely: SSD [6], YOLOV2 [7], and faster R-CNN [8] to classify traffic vehicles. The authors tested their approach on videos captured from different sites and demonstrated that they had achieved high accuracy and fast processing speed. The research in [9] proposed a semi-supervised vehicle type classification approach based on a broad learning scheme (BLS) [10] to reduce the cost of the training phase, and a dynamic ensemble structure is used to estimate the class type probabilities. Naghmeh et al. [11] proposed a semi-supervised Fuzzy C-Means clustering to predict different vehicle types and labels. They utilized unlabeled and labeled data to extract useful information for classifying the vehicles. Additionally, random oversampling was used to handle the multi-class imbalanced datasets. Semi-supervised principal component analysis convolutional network (PCN) was adopted in [12]. The authors generated a convolutional filter bank to leverage the effectiveness of traditional convolution neural networks against different image schemes, including translation, scale, noise, and rotations. The research incorporated PCN into softmax and support vector machine (SVM) classifiers to evaluate different structures.

Nadiya et al. [13] proposed a monocular vision-based technique to detect vehicles for automatic toll collection. Their technique suggests using a group of convolutional neural networks to estimate the probability for vehicles class and feed them into a gradient-boosting classifier to complement the labels obtained from optical sensors. The research in [14] introduced an automated approach for different vehicle classifications from rear-view images by fusing physical and spatial attributes. The physical features include height from the ground to the rear bumper and the distance from the bumper to the license plate. The spatial features are extracted by applying a convolutional neural network. The SVM classifier is employed to classify fused features.

Unsupervised Feature Learning was discussed and elaborated by Amir et al. [15]. The approach depends upon the generation of a dictionary from a dense scale-invariant feature extractor [16]. Bases are generated by applying the k-means clustering technique. The generated dictionary maps the input features to a new learned space by employing a coding vector with the trained bases. In classifying images, they utilized the spatial pyramid matching [17] and SVM classifier. Zhanyu et al. [18] introduced a fine-grained classification scheme based on convolutional neural networks. They added a new max-pooling layer between the fully connected and traditional layers. The proposed layer reduces the dimensions of feature space, and the computational cost is reduced.

Zhang et al. [19] utilized single-shot multi-box feature detectors based on deconvolution and pooling to classify vehicles using convolutional neural networks. The authors in [20] compared the performance of vehicle detection and classification using the mixture of Gaussian (MoG) background subtraction algorithm along with a support vector machine (SVM) vehicle classification model against the faster RCNN. Their experiments showed that faster R-CNN outperforms the SVM method. Xiang et al. [21] proposed a method to count vehicles based on a moving-object detector that works with static and moving backgrounds. A pixel-level detector handles static objects, while camera registration is employed with moving backgrounds. In their approach, an online learning tracker counts vehicles under different situations. The authors in [22] introduced an approach to identify vehicle make and model from front-facing images utilizing physical and visual characteristics. First, they extracted the logo and fed it into a classifier to identify the vehicle make. Then, they customized a hierarchical classifier to identify the vehicle type.

Farahani [23] utilized the Gaussian mixture model for background segmentation, and the extended mean shift (EMS) algorithm based on color information was utilized for tracking. The EMS algorithm was adapted by applying an Epanechnikov kernel estimator function on the feature space. This feature space was derived from the color histogram. Roy et al. [24] extracted foreground objects using GMM and proposed a new weighted mean approach to track vehicles. Das et al. [25] classified vehicles into five classes using a bag of Speeded Up Robust Features (SURF). An SVM classifier was adopted in their work. The authors in [26] suggested two-stage methods to classify vehicles. Firstly, they balanced the data samples by augmenting the acquired data. Secondly, the augmented data was classified utilizing an ensemble of convolutional neural networks (CNN) with different architectures and learned on the augmented training dataset. Cai et al. [27] pioneered a scene-adjusted vehicle detection approach based on the concept of the bagging (bootstrap aggregating) technique. A composite deep-structure classifier was built using multiple classifiers of samples generated in the target scene based on a confidence score with a voting mechanism. The authors in [28] adopted latent SVM in a vehicle make and model recognition. A novel greedy parts localization algorithm was employed to extract some descriptive parts in the vehicles used in the learning stages. Chavez-Garcia et al. [29] located vehicles within a video frame using a two-dimensional Bayesian occupancy grid map. They experimented with a sparse version of the Histogram of Oriented Gradients (HOG) along with the Adaboost classifier in the classification stage. The authors in [30] detected vehicle candidates utilizing the active basis model (ABM) based on Gabor wavelet elements. A random forest classifier is trained to classify vehicles into three categories based on the vehicle length extracted from the spatial domain image and the gray-level co-occurrence matrix of the detected vehicle image. Siddiqui et al. [31] utilized a bag of SURF features of either the front- or rear-facing images of vehicles in conjunction with multi-class support vector machine (SVM) classifier and attribute-bagging-based ensemble of SVM (AB-SVM) to recognize vehicle make and model.

Toropov et al. [32] employed adaptive GMM for segmenting vehicles and Viola–Jones cascade detector based on Haar features for vehicle tracking. Vehicles are counted using a probabilistic counting model. The authors in [33] provided a method for detecting and tracking vehicles in videos captured from roadside cameras. Authors in [34] proposed an integrated work of vehicle detection, tracking, and classification for purposes of emission estimation. Similarly, the work in [35] provides a detection and tracking algorithm for on-road vehicles using a car-mounted camera. Vehicle detection and tracking are extended to include vehicle counting in [36] and vehicle lane assignment and analysis in [37]. The authors in [38,39] used classifiers to distinguish between moving objects that are considered cars as opposed to another type of vehicle. Similarly, classification was used to distinguish trucks from other highway traffic for a study on a truck and industrial vehicle density in [40]. Techniques that apply multiclass classification have also been used to distinguish between vehicle categories on the highway, such as cars, buses, motorcycles, and trucks [41,42,43]. The work in [44] distinguishes multiple vehicles by size to assign correct toll rates.

When analyzing the previously discussed efforts, it is clear that a trend of strategies for completing a particular purpose has been developed. These efforts can be categorized into detection approaches, known as foreground segmentation, tracking algorithms, and (if applicable) feature extraction and vehicle categorization methods. Furthermore, some of the previously summarized approaches examined techniques utilizing enhancements in ROI extraction and camera alignment. While deep learning (DL) is a fast-growing area of research in vehicle segmentation for intelligent transportation systems (ITS), these methods need a lot of data to train. Thus, they cannot function properly without huge training data and imbalanced datasets. Additionally, optimizing DL architecture parameters requires plenty of experiments using deep learning networks and architecture combinations. The dataset utilized in this work is small and is not enough to adopt a deep learning approach [15,45,46,47,48,49,50]. The results section compares the proposed technique for verification and performance evaluation against recent deep learning architectures with different deep neural networks.

In this work, the authors develop and implement a monocular camera viewpoint-independent and fully integrated method for vehicular traffic segmentation and classification. Key results reported in this work are evaluated using networked traffic camera videos obtained from the Virginia Department of Transportation (VDOT). More specifically, the following contributions are presented: (1) an automated region of interest (ROI) extraction approach; (2) an improved real-time foreground segmentation algorithm; (3) a classification scheme that is capable of sensing multiple vehicle clusters for handling occluded vehicles in dense traffic scenarios; and (4) performance evaluation of twenty-three object detection deep learning architectures that have been trained and adapted to the evaluated dataset employing transfer learning.

The rest of this paper is organized as follows: Section 2 details the proposed system pipeline. Then, Section 3 presents the results of the developed system where the performance of different system stages is evaluated to accurately segment individual vehicles and to classify the vehicles by type. Finally, in Section 4, concluding statements are discussed.

## 2. Proposed Technique

Deep learning approaches combine both feature extraction and classification into a single procedure. They require massive training datasets to achieve high accuracy. Unlike deep learning approaches, traditional supervised machine-learning approaches have reported high accuracy in object detection and classification with limited and small amounts of data. These approaches require two substages: one for feature extraction and the other one for object classification. The histogram of oriented gradients approach has been widely used in object detection based on the gradients of visual and textural differences with reported high detection accuracy. Additionally, morphological, size, and shape features are utilized to separate various classes representing different objects. The proposed technique utilized the support vector machine classifier since it has been known for high classification accuracy in similar domains and requires few training samples compared to deep learning approaches.

This section develops methods to solve the primary stated goal of autonomous vehicle segmentation and type classification in a traffic video database. The developed pipeline consists of the following steps: (1) automated development of an ROI; (2) foreground object segmentation using improved robust low-rank and sparse matrix decomposition; (3) extraction of vehicle size, shape, and texture features using morphological properties and histogram of oriented gradients (HOG); (4) vehicle tracking using the extended Kalman filter (EKF) method; (5) vehicle classification using a multiclass support vector machine (SVM) classifier; and finally, (6) multiple vehicle handling using a K-means over-segmentation and a two-pass classification scheme. The proposed pipeline is shown in Figure 1, and the following subsections detail its steps.

### 2.1. Region of Interest Extraction

The proposed algorithm’s first step establishes an ROI in the image scene to reduce the computational complexity, accelerate processing, and retain the most visual features of the captured frames. Most practical traffic surveillance cameras are set at a viewing angle to monitor a large section of the highway. As vehicles progress through the image scene toward the horizon and move farther from the camera, the vehicle size becomes smaller and can appear as image noise. Likewise, as vehicles become smaller, there are not enough meaningful details to extract features for classification purposes, and detected vehicles at a far distance can interfere as outliers for training a classifier. In that regard, the processed frame is clipped to 70% of its height, considering the camera side. In this proposed vehicle detection algorithm, monitoring is optimized to only handle vehicles moving along one major angle. An automated ROI extraction approach is introduced to define the effective part of the image scene in which vehicles can be efficiently segmented and classified. The ROI is a binary mask applied to each video frame to filter pixel intensities outside the ROI and set them to zero.

The proposed extraction of ROI includes a sequence of six steps, namely: (1) noise suppression, (2) edge detection, (3) Hough transform, (4) fuzzy C-means, (5) curve fitting, and (6) mask development. The pipeline for the ROI extraction approach is depicted in Figure 2.

Since the captured frames are noisy and degraded and have artifacts due to environmental surroundings, a Gaussian kernel is utilized to suppress these artifacts and to keep strong edges representing potential highway lane sides. A canny edge detector is employed to detect candidate edges from blurred images, followed by connected components analysis to suppress unwanted edges in small areas. Then, the Hough transform is applied to detect possible lines in the edge image. Vertical lines whose angles are less than $2^{o}$ are suppressed. It is assumed that the possible lanes will not be vertical lines since these vertical lines may account for other objects within the scene, such as light poles. Applying the Hough transform to the image results in a feature space of two parameters, namely, ρ and θ, where ρ represents the normal from the origin to the detected lines, while θ represents the inclination to *x*-axis. Next, fuzzy C-mean is applied to the Hough space features to cluster the possible detected Hough lines using the two features, ρ and θ. The number of clusters is set empirically to four clusters. Since the detected clustered lines may be represented as broken segments near each other, curve fitting is employed to fit a cluster of detected broken segments into a unique single line. Several 1D curve fitting techniques are tested, including linear and higher order polynomials, exponential, Gaussian, and cubic spline models. Empirically, it is found that the linear polynomial model provides the best results. Finally, a raster scan is utilized across the image overlaid by fitted curves. These curves are used as guidelines to define the masked areas.

### 2.2. Foreground Segmentation

The foreground segmentation process identifies moving vehicles through the highway image scene. This stage is a crucial step in the proposed algorithm since it is required to extract vehicles with high accuracy and low computational time to make the system suitable for real-time implementations. Plenty of approaches have been developed in the literature related to foreground segmentation. Of these approaches are the robust low-rank and sparse matrix decomposition in which a low-rank matrix represents the background image, and a sparse matrix represents the foreground objects [51]. These techniques prove to achieve high accuracy and real-time implementations. Sobral et al. [52] have reported several low-rank decomposition algorithms for background extraction from videos, and they ranked those algorithms in terms of their computational time.

In this research, the authors first investigated several approaches and applied them to the available dataset, which is noisy, degraded highway videos captured under different weather conditions. The evaluation is in terms of segmentation efficiency and computational time. Secondly, the proposed algorithm improves upon one of the evaluated algorithms named fast principal component pursuit (FPCP) introduced by Rodriguez and Wohlberg [53]. The proposed approach improves the speed and segmentation accuracy and verifies the improved performance using a manually labeled dataset. The proposed algorithm utilizes the FPCP approach as an initial stage to pre-train the system. The pre-training provides a priori low-rank background representation that is used in subsequent stages. Once a low-rank representation is determined, a simple frame difference is utilized to extract the sparse foreground images. Another aspect is that FPCP and other robust low-rank and sparse matrix decomposition approaches utilize global thresholding to extract foreground objects. However, global thresholding extracts most of the segmented objects and labels them as foreground objects; it mislabels other foreground objects, especially when the pixel values of these objects are close to background pixel intensities. Additionally, frame difference provides negative values considered background objects when global thresholding is applied; however, they are not. Figure 3 depicts the histogram of sparse frames that results from the difference between estimated low-rank background frames and a frame under processing. This gives intuition to fit the resultant sparse matrix into a Gaussian distribution with a mean representing the background pixel values. If values fall in the Gaussian distribution outer sides, they are mapped to the segmented foreground pixels.

In that sense, the proposed algorithm models the resultant difference matrix into a normal distribution and determines the threshold utilizing the fitted normal distribution mean and variance. The estimated mean is around or close to zero values. Those values represent the background pixel intensities. Considering that noises are present in the scenes, the background pixel values are mapped inside the interval defined by the bounds μ±λσ. In contrast, foreground pixel values are mapped outside that interval.

Following the same FPCP formulation, a data matrix *X* can be written as X=L+Z such that *L* is a low-rank matrix, and *Z* is a sparse matrix satisfying the following optimization problem:(1)$$\underset{L,Z}{\arg\min}{∥L∥}_{*}+\lambda {∥Z∥}_{1}\;s.t.\;X=L+Z$$
where $X\in R^{m\times n}$ is the data matrix, ${∥•∥}_{*}$ is the nuclear norm, while ${∥•∥}_{1}$ is the $l^{1}$ norm of a given matrix. The data matrix *X* is constructed by stacking the video frames shaped as columns of *X*. The FPCP authors relaxed the nuclear norm, and the optimization function became:(2)$$\underset{L,Z}{\arg\min}\frac{1}{2}{∥L+Z-X∥}_{F}+\lambda {∥Z∥}_{1}\;s.t.\;\mathrm{rank}\left(L\right)=t$$
where ${∥•∥}_{F}$ is the Frobenius norm of the matrix, and *t* represents the number of partially selected components of the spectral value decomposition (SVD). The FPCP approach proposed a solution to Equation (Equation 2) using alternating minimization as follows:(3)$$L_{k+1}=\underset{L}{\arg\min}{∥L+Z_{k}-X∥}_{F}\;s.t.\;\mathrm{rank}\left(L\right)=t$$
(4)$$Z_{k+1}=\underset{L}{\arg\min}{∥L_{k+1}+Z-X∥}_{F}+\lambda {∥Z∥}_{1}$$

Equation (Equation 3) is addressed utilizing partial *t* components of the SVD of X−Z, while Equation (Equation 4) is solved by soft thresholding $\mathrm{shrink}(X-L_{k+1},\lambda )$ where the soft thresholding is defined by:(5)shrink(x,ϵ)=sign(x)max{0,|x|−ϵ}

The videos’ set of *k* frames are used to pre-train the system to estimate a low-rank matrix $\tilde{L}$ that represents the background. A simple frame difference between the estimated background $\tilde{L}$ and the current video frame $F^{\left(i\right)}$ is used to determine the current spare matrix that represents the foreground, $Z^{\left(i\right)}$, as given by:(6)$$Z^{\left(i\right)}=F^{\left(i\right)}-\tilde{L}$$

Once $Z^{\left(i\right)}$ is computed, resultant difference values are fitted into a normal distribution of mean μ and standard deviation σ. The minimum variance unbiased estimator (MVUE) is utilized to estimate the parameters μ and σ. The sample mean, $\bar{x}$, is defined as
(7)$$\bar{x}=\sum_{i=1}^{n}\frac{x_{i}}{n}$$
while the sample variance, $s^{2}$, is defined as
(8)$$s^{2}=\frac{1}{n-1}\sum_{i=1}^{n}{\left(x_{i}-\bar{x}\right)}^{2}$$

The sample mean and variance are used to estimate the mean and standard deviation, respectively. Next, the foreground is segmented utilizing the following threshold:(9)$$Z^{\left(i\right)}\left(x,y\right)=\left\{\begin{array}{cc} 0 & \mathrm{if}\;Z^{\left(i\right)}\left(x,y\right)\in \mu \pm \lambda \sigma \\ 1 & \mathrm{otherwise} \end{array}\right.$$
where λ is determined empirically from simulations, and (x,y) are spatial coordinates of the pixel values within the frame $Z^{\left(i\right)}$. Finally, binary opening and closing operations are integrated to reduce the noise effects after thresholding.

### 2.3. Vehicle Features

The authors utilize the same approach presented in their previous work [54] to extract segmented vehicle features. Size, shape, and texture descriptors are considered for classification. Morphological properties are used to describe the size and shape of each vehicle. Gradient-based texture features are calculated using a HOG algorithm on the rectangular boundary image of the detected vehicle.

Twelve extracted feature descriptors representing the size and shape of segmented vehicles are employed. These descriptors include area, bounding box, centroid, convex area, eccentricity, equivalent diameter, Euler number, extent, primary axis length, minor axis length, orientation, and perimeter [54,55].

In addition to the physical features, textural features are integrated by computing the histogram of oriented gradients of segmented vehicles. Due to its effectiveness in identifying an object’s visual texture, the HOG descriptor is widely used in image processing and computer vision for object detection [56]. The HOG features are evaluated using both image gradient magnitudes and orientations and are slightly affected by local geometric and lighting distortions.

### 2.4. Vehicle Tracking

In order to keep an accurate count of vehicles within the highway image scene, each detected vehicle is tracked as it traverses the ROI. A Kalman filter tracking approach is used to track each detected vehicle through the ROI [57]. The Kalman filter vector is created for each newly detected vehicle. It tracks the ID number of the vehicle, the centroid position, and the vehicle velocity assuming constant acceleration through the ROI. Image features derived from the foreground-segmented vehicle are also paired with the Kalman filter track for every frame in which the vehicle is present. Once the vehicle exits the ROI, the Kalman filter vehicle track is deleted.

### 2.5. Classification

Once a vehicle track has been deleted, as described in the previous section, the classification of the vehicle at every frame occurs. The classification of vehicles occurs in three steps: (1) training the classifier, (2) multiclass SVM classification, and (3) statistical analysis for final classification.

#### 2.5.1. Training the Classifier

The classifier is trained using a manually labeled ground truth dataset for each camera. Eight classes are considered for training. These classes include the six vehicle classes: passenger car, passenger truck, bus, small utility truck, and large utility truck. In addition, classes representing a detection that is not a vehicle or a detection containing multiple vehicles are presented. The classifier is trained on a manually tagged dataset using 10-fold cross-validation in which the dataset is divided randomly into 10 groups. The training is conducted using nine groups; the remaining group is used for testing. The process is repeated ten times to estimate the classifier’s performance.

#### 2.5.2. Multiclass SVM Classification

Multi-class SVM classifier is adopted to classify the segmented data into one of eight classes. The SVM classifier is known to be efficient in separating large feature vectors and does not require intensive memory allocations [58]. A set of binary SVM classifiers is adopted to handle the multi-class separation. The SVM classifier with radial basis function (RBF) kernel is chosen for its better separation of nonlinear feature vectors. The RBF kernel has two parameters, called the sigma and box constraint, found by a grid search method. In the grid search method, each parameter varies over a range of 0.1 to 15, and the 10-fold cross-validation is performed. The parameters are set for the maximum cross-validation result.

Following the same approach discussed by Chen et al. [59], the work presented here adopts eight binary SVM classifiers to handle the problem of multi-object classification. First, one SVM classifier is trained to separate a certain positive class from other classes treated as a single negative class. Then, after the training phase of the chosen classifier is over, the same process continues for training other classifiers.

#### 2.5.3. Final Classification Result

The final classification result is found by taking the statistical mode of each frame classification result. For example, a single vehicle track may make a mistake in tracking and correct to the wrong detected vehicle. Furthermore, the tracked vehicle may also include partial detection at the ROI boundary and be classified as “Not a Vehicle”. Using the mode of the classification results eliminates these conditions and uses the most stable classification results from each frame in the vehicle track.

### 2.6. Multiple Vehicle Handling

Due to vehicular occlusion, some vehicles traveling nearby are clustered and appear as a single vehicle. The classification method in the previous step is trained to recognize multiple vehicle clusters. If the deleted track contains multiple vehicles, then occlusion is assumed. This occlusion is handled through an iterative over-segmentation and reclassification method.

#### 2.6.1. K-Means Over-Segmentation

The first step in over-segmentation is to analyze the geometry of the boundary rectangle for the occluded vehicles. If the boundary rectangle width is greater than the height, it is assumed that the multiple vehicles are arranged from left to right, and the boundary rectangle is divided vertically. Otherwise, it is assumed that the multiple vehicles are arranged from top to bottom, and the boundary rectangle is divided horizontally. Two centroids are assumed in the newly divided boundary rectangle.

K-means clustering is used to perform over-segmentation of the multiple-vehicle cluster. K-means uses the image intensity and quantizes similar intensity levels into a single patch. This over-segmentation process uses up to 25 levels applied to the input image of each multiple-vehicle cluster.

After the newly segmented patches are created, the centroid of each patch is calculated. The minimum Euclidean distance of each patch to the new boundary box centroids is used to assign each patch to the new boundary box. Once the patches are assigned, the newly clustered vehicle is converted from the patches. Features are extracted from the new vehicle cluster.

#### 2.6.2. Reclassification

The final step in multiple vehicle handling is to perform classification on the two detected vehicles from the k-means over-segmentation process. This reclassification follows the method described in Section 2.5. The steps in this section are repeated if the vehicle cluster still contains multiple vehicles. If the vehicles cannot be separated, this iterative approach will only repeat three times.

## 3. Results and Discussion

A manually labeled ground truth dataset was created for evaluation. Four evaluated sample videos are described in Table 1. These videos were manually labeled for vehicle segmentation and classification. Each video shows a segment of an interstate under varying conditions of traffic and environmental scenarios. Figure 4 shows a sample of vehicles within the given dataset. It can be seen that the given dataset is poor in its spatial resolution, and its visual quality content is highly degraded and under-sampled. The size of bounding boxes containing the extracted vehicles is 40 by 30 pixels extracted from video frames of dimensions 320 by 240 pixels at a spatial resolution of 72 dots per inch (dpi).

### 3.1. ROI Extraction

The proposed ROI extraction technique described in Section 2.1 is applied to the first frame of the collected videos to prepare and feed the extracted ROI mask to subsequent processing of other frames. The processed frame is filtered utilizing a Gaussian blur filter with a 3×3 window. A canny edge detector is adopted to detect the edges from the blurred image. The Hough transform is applied, and empirically ten peaks are selected. Hough lines are extracted with a minimum length of 20. FCM clustering algorithm is applied with four clusters. Finally, linear polynomial fit is used to fit the clustered line segments into lane sides. Detailed results of different ROI pipeline stages are depicted in Figure 5 and Figure 6 for two different sample videos. The proposed technique demonstrates good results in the extraction of ROI. It works well for line-shaped lanes. While the performance has a small degradation for curvy lanes, as seen in Figure 5f, it is still acceptable since the scope of this approach is not for lane departure systems that require precise extraction of the ROI. Additionally, applying the Gaussian kernel to images reduces the amount of extracted edges, helps to identify the strong ones, and reduces the not-required Hough lines. FCM works to group different line segments nearby into unique cluster lines. Finally, the linear curve fitting approximates the clustered lines into a single line representing highway lanes’ sides. The ROI extraction results in an almost 40% reduction of pixels processed by the algorithm’s subsequent stages. The overall computational time is reduced due to applying an ROI mask.

### 3.2. Foreground Segmentation Evaluation

The proposed algorithm utilizes the FPCP approach as an initial stage to pre-train the system. First, a set of frames determined empirically are utilized as initialization to generate a priori low-rank background representation used in subsequent frame differentiating stages. Then, the frames are reshaped into column vectors stacked side-by-side to construct a matrix fed to the FPCP algorithm to estimate a background frame. Utilizing 60 frames in the training phase reduces the processed data by almost 80% for Videos 1, 2, and 3 and 94% for Video 4. Once the frames are subtracted from the pre-trained low-rank matrix, the differentiated frames are fitted into the Gaussian distribution. Then, the mean and variance are estimated from the extracted difference sparse matrix. Thresholding is applied utilizing Equation (Equation 9) in which the tuning parameter λ is empirically set to five. Foreground vehicle segmentation is evaluated using classification accuracy, precision, recall, and F-measure. Classification accuracy (CA) is defined as
(10)CA=(TP+TN)/(TP+FN+FP+TN)
where *TP* is the true positive, *TN* is the true negative, *FP* is the false positive, and *FN* is the false negative. Since classification accuracy is not a good representative for evaluating algorithms when the available dataset is imbalanced, it has been advised through the research communities to use the F-measure as a good metric to evaluate the performance of segmentation and detection techniques. The F-measure is determined by:(11)$$F-\mathrm{meaure}=2*\frac{\mathrm{Precision}*\mathrm{Recall}}{\mathrm{Precision}+\mathrm{Recall}}$$
where Precision is defines as:(12)$$\mathrm{Precision}=\frac{TP}{TP+FP},$$
and Recall is defined as:(13)$$\mathrm{Recall}=\frac{Tp}{TP+FN}$$

*TP*, *TN*, *FP*, and *FN* detected pixels are determined by comparing the algorithm outputs to a manually labeled ground truth for the given dataset. Moreover, the proposed algorithms are compared against recent foreground segmentation techniques including: fast principal component pursuit via alternating minimization (FPCP) [53], Grassmann averages for scalable robust PCA (GA) [60], low-rank and sparse matrix decomposition via the truncated nuclear norm and a sparse regularizer (LRSMDTNNorm) [61], shifted subspaces tracking on sparse outlier for motion segmentation (GreGoDec) [62], robust principal component pursuit via inexact alternating minimization on matrix manifolds (RPCP ) [63], scalable robust matrix recovery: Frank–Wolfe meets proximal methods (FW-T) [64], robust principal component analysis with complex noise (MoG-RPCA) [65], a variational approach to stable principal component pursuit (Lag-SPCP-QN) [66], and robust PCA via nonconvex rank approximation (NonConvRPCA ) [67]. The previously mentioned classification metrics and the average computational time per frame (Ave comp. time) are used to compare the proposed technique performance against those algorithms.

Table 2, Table 3, Table 4 and Table 5 list different evaluation metrics for the proposed algorithm against other segmentation algorithms for four different videos. For Video 1, which represents a sample of sunny daytime traffic, the proposed algorithm showed an improvement of 10% in terms of the F-measure, and it is 20% faster than the GA approach, which is the fastest technique among the compared algorithms. A sample of traffic with low light and dusk conditions is represented by Video 2. Improvements of about 20% in the F-measure and 80% in the computational speed are achieved. Rainy and daytime traffic condition is depicted in Video 3. However, the proposed algorithm improved by 14% in the F-measure and 83% in the computational time; nevertheless, the quality of the segmentation process needs to be improved. Since Video 3 represents rainy conditions, the proposed algorithm and other evaluated algorithms failed to deal effectively with reflections due to rain that are considered moving objects and consequently interpreted as foreground objects. Performance improvements for Video 4 are 12% and 36% in terms of F-score and computational time, respectively. Figure 7, Figure 8, Figure 9 and Figure 10 depict the visual quality of segmented vehicles for the proposed technique against other evaluated algorithms. It can be seen that the proposed algorithm is better than other techniques, and more accurate segmentation of vehicles for different videos has been achieved. Techniques GA, Lag-SPCP-QN, and GM have poor showing for the different evaluated videos.

### 3.3. Classification Evaluation

We used 10-fold cross-validation to train the classifier compared to a manually labeled dataset. Evaluation metrics, including precision, recall, and F-score, are utilized to evaluate the classification performance for each class. A ROC plot is given for each class considered for classification. The confusion matrix for each classification result is also obtained.

#### 3.3.1. Video 1

Video 1 has a total of 1008 vehicle detections. These detections include 523 passenger cars, 257 passenger trucks, 129 multiple vehicle detections, and 99 non-vehicle detections. The grid search result gives a box constraint of 11 and a sigma of 4 for the RBF parameters. The confusion matrix is shown in Table 6, while classification evaluation metrics are shown in Table 7. The overall accuracy for Video 1 is 86.71%. Regarding the F-score, non-vehicle objects classification performance is the worst among different classes. There is excellent handling for multiple vehicle cluster detection and passenger car classes. Figure 11 shows the ROC curves with an average area under the curve (AUC) of 0.8963 for all classes.

There are 123 predicted multiple vehicle clusters detected in Video 1, representing 264 total vehicles that are a subset of the final classification result. The confusion matrix for detected vehicles from multiple vehicle clustering algorithm output is described in Table 8 with an overall accuracy of 72.73%. These 264 vehicles are omitted from the output data without the multiple vehicle cluster handling. Additionally, classification evaluation metrics for multiple vehicle clusters are presented in Table 9. The algorithm performs well in identifying various vehicle classes from clustered objects.

#### 3.3.2. Video 2

Video 2 has a total of 373 vehicle detections. These detections include 120 passenger cars, 86 passenger trucks, 13 small utility trucks, 14 multiple vehicle detections, and 140 non-vehicle detections. The grid search result gives a box constraint of seven and a sigma of three for the RBF parameters. The confusion matrix is depicted in Table 10, while classification evaluation metrics are shown in Table 11. The overall accuracy for Video 2 is 84.72%. Figure 12 shows the ROC curves with an average AUC of 0.9088 for all classes. In terms of the F-score metric, there is an excellent performance for different classes classification.

There are 12 predicted multiple vehicle clusters detected in Video 2, representing 26 total vehicles that are a subset of the final classification result. The confusion matrix for detected vehicles from multiple vehicle clustering algorithm output is described in Table 12 with an overall accuracy of 66.67%. Additionally, classification evaluation metrics for multiple vehicle clusters are presented in Table 13. The algorithm performs well for both passenger car and truck classes, while it has abysmal performance for small utility trucks. This is because the small utility truck has a poor showing in the dataset, and consequently, the algorithm failed to identify the class without enough training data.

#### 3.3.3. Video 3

Video 3 has a total of 1948 vehicle detections. These detections include 464 passenger cars, 478 passenger trucks, 67 large utility trucks, 189 multiple vehicle detections, and 744 non-vehicle detections. The grid search result gives a box constraint of 11 and a sigma of 3 for the RBF parameters. The confusion matrix is shown in Table 14, while classification evaluation metrics are shown in Table 15. The overall accuracy for Video 3 is 78.75.71%. Regarding the F-score, the algorithm performs extremely poorly classifying motorcycles, buses, and small utility trucks. This is expected since there is not enough training data representing these classes. Figure 13 shows the ROC curves with an average AUC of 0.8666 for all classes.

There are 237 predicted multiple vehicle clusters detected in Video 3, representing 546 total vehicles that are a subset of the final classification result. The confusion matrix for detected vehicles from multiple vehicle clustering algorithm performance is described in Table 16 with an overall accuracy of 61.36%. Additionally, classification evaluation metrics for multiple vehicle clusters are presented in Table 17. The algorithm performs well in identifying both passenger car and truck classes, while it has poor performance for other classes.

#### 3.3.4. Video 4

Video 4 has a total of 2868 vehicle detections. These detections include 592 passenger cars, 671 passenger trucks, 22 buses, 125 small utility trucks, 58 large utility trucks, 269 multiple-vehicle detections, and 1131 non-vehicle detections. The grid search result gives a box constraint of 11 and a sigma of 3 for the RBF parameters. The confusion matrix is shown in Table 18, while classification evaluation metrics are shown in Table 19. The overall accuracy for Video 4 is 86.37%. The algorithm shows good performance in identifying various classes. Figure 14 shows the ROC curves with an average AUC of 0.8984 for all classes.

There are 234 predicted multiple vehicle clusters detected in Video 4, representing 513 total vehicles that are a subset of the final classification result. The multiple vehicle clustering algorithm performance is described in Table 20 with an overall classification accuracy of 71.92%. Additionally, classification evaluation metrics for multiple vehicle clusters are presented in Table 21. The algorithm shows poor performance in identifying both bus and large utility truck classes.

### 3.4. Comparison with Deep Learning Architectures

While deep learning is a fast-growing area of research in different domains of image and video processing and computer vision, it has several drawbacks. Different tactics have been tried in intelligent transportation systems (ITS). Favorable and accurate results have been demonstrated compared to traditional methods, but it requires machines with intensive memory storage and huge computational power. Adding to that, it requires a huge amount of training data. Additionally, its configuration and tuning are a hard and lengthy process. Optimizing the deep learning architecture parameters requires too many experiments running for long periods to set the optimal settings [45,46,47].

The work presented in this paper evaluates the performance of twenty-three different deep learning architectures for object detection that have been previously trained on either the COCO dataset or ImageNet. Since the training process needs high computational resources and graphical processing units (GPUs) and the process is time-consuming, transfer learning is adapted in this work [68]. The different evaluated models utilize the weights from pre-trained networks to fine-tune these architectures on new datasets. The evaluated deep object detection architectures are: Faster R-CNN [8], YOLO V2 [69], YOLO V3 [7], and YOLO V4 [70]) and the feature extractor networks are (Resnet18, Resnet50, Resent101 [71], inception V3 [72], inceptionresnet V2 [73], darknet53 [74], squeezenet [75], and googlenet [76]). The combinations of these architectures are evaluated in terms of the average precision, which is defined as the area under the curve representing the relation between the recall and precision at different thresholds. The performance of these deep learning models is evaluated on small datasets using transfer learning and is compared against the proposed traditional technique.

The trails of deep learning architecture and feature extractors networks evaluated in this work are listed in Table 22. As depicted from the table, not all possible combinations were examined. This is due to the lack of high computational power and memory. Additionally, many experiments were carried out for each combination to fine-tune the network parameters.

The dataset was divided into 60% for training, 10% for validation, and 30% for testing the trained detector. Data augmentation was applied to the training data to improve the training and increase the number of labeled data. The data augmentation included color jitter in HSV space, random scaling, and random horizontal flip.

Table 23 lists the average precision of the evaluated deep learning combinations against the proposed algorithm. It can be seen that the performance of deep learning architectures was abysmal. This is expected since the available dataset is small and does not provide enough labeled training samples to train the deep learning networks. The proposed algorithm improved the performance against the evaluated algorithms, especially for Videos 1, 2, and 4. However, Video 3, representing rainy conditions, had the worst classification performance among different algorithms.

### 3.5. Timing Analysis

The algorithm was tested on a computer running as a single thread on 11th Gen Intel(R) Core(TM) running at 2.80 GHz with 16 GB of RAM with no GPU card installed. The timing analysis, given in Table 24, summarizes the time required to run each sample video through the algorithm and the average frame rate. The timing analysis shows that the approach implementation can operate in real time at 15 frames per second.

## 4. Conclusions

In this work, an end-to-end integrated vision-based framework that is capable of segmenting moving vehicles and classifying them by type is presented. Novel contributions to intelligent transportation systems research have been presented by an integrated algorithm capable of vehicle segmentation and classification in a camera-viewpoint-independent highway image scene with the ability to handle multiple vehicle clusters. The algorithm was speeded up through an automated approach for extracting a region of interest, reducing 40% in the processed data. Vehicle segmentation was carried out using an improved speeded-up robust low-rank matrix decomposition technique. Utilizing a new and effective thresholding method improved segmentation accuracy and simultaneously sped up the segmentation processing. It improved segmentation accuracy by 15% on average compared to current approaches and was 55% quicker on average than recent segmentation techniques. Segmentation was shown to perform very well during the day. However, the segmentation and classification results degraded under low light conditions or poor environmental scenarios such as rain.

The algorithm adopted size and shape physical descriptors from morphological properties and textural features from the histogram of oriented gradients (HOG) extracted from the segmented traffic for feature extraction. Moreover, a multi-class support vector machine classifier was used to identify diverse traffic vehicle categories, such as passenger cars, passenger trucks, motorbikes, buses, and small and big utility vehicles. The proposed approach manages multiple vehicle clusters through an iterative k-means clustering over-segmentation procedure. It shows high classification performance for vehicle types, such as passenger cars and passenger trucks, while its performance was disappointing for classes with a poor showing in the training dataset. Additionally, the algorithm was compared against 23 different deep-learning architectures. The resulting algorithm outperformed the compared deep learning algorithms for the quality of vehicle classification accuracy.

The timing analysis results show that the algorithm’s current implementation can perform in real time for the 15 frames per second frame rate. However, future modifications to the algorithm to handle different weather scenarios, such as rainy conditions, can improve the segmentation and classification of the proposed algorithm and make it adaptable to different weather conditions. Additionally, the lack of enough training datasets close to the one utilized in this research makes using deep learning techniques not promising and challenging and leads to poor performance.

## Figures and Tables

**Figure 1 sensors-22-08121-f001:**
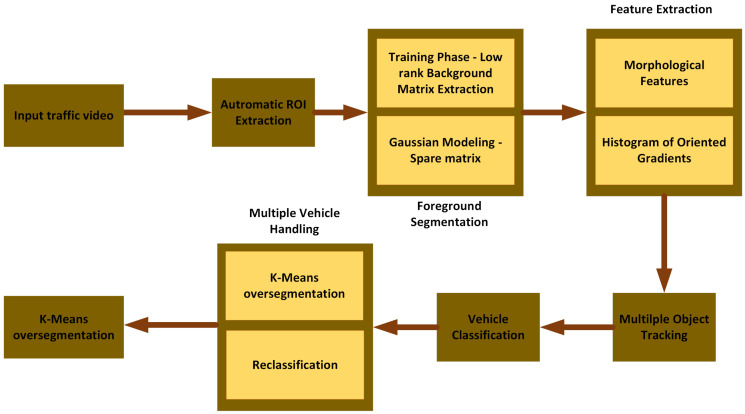
Proposed algorithm flow diagram.

**Figure 2 sensors-22-08121-f002:**

ROI extraction pipeline.

**Figure 3 sensors-22-08121-f003:**
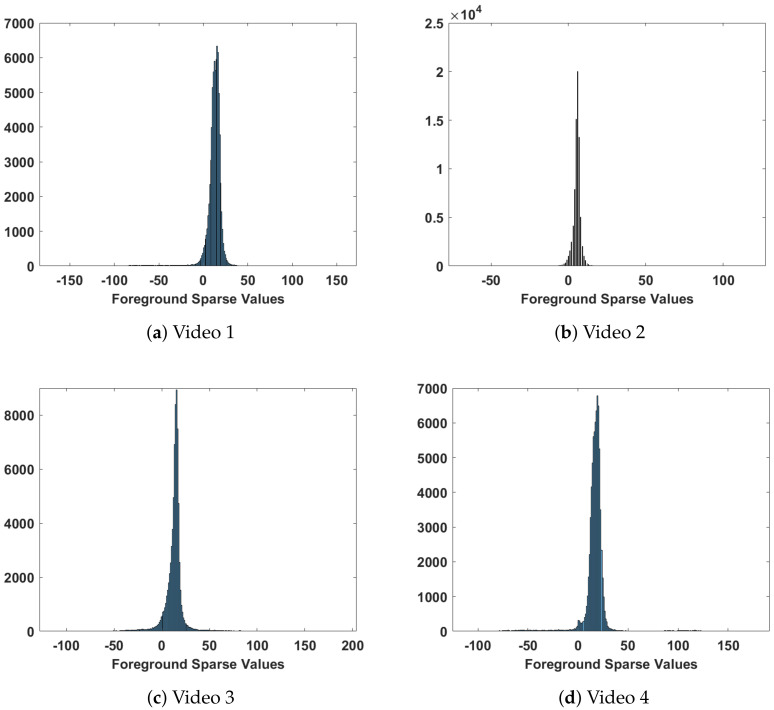
A sample histogram of extracted sparse background for different dataset videos.

**Figure 4 sensors-22-08121-f004:**
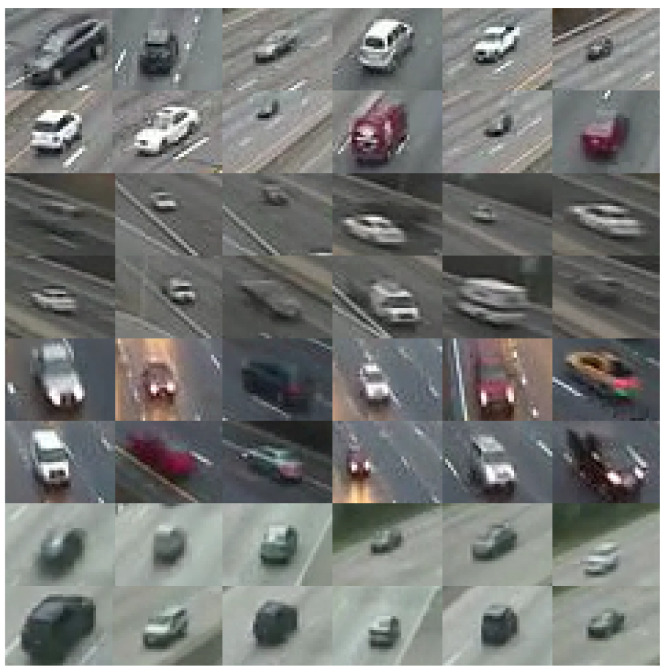
Sample vehicles extracted from different videos in the dataset.

**Figure 5 sensors-22-08121-f005:**
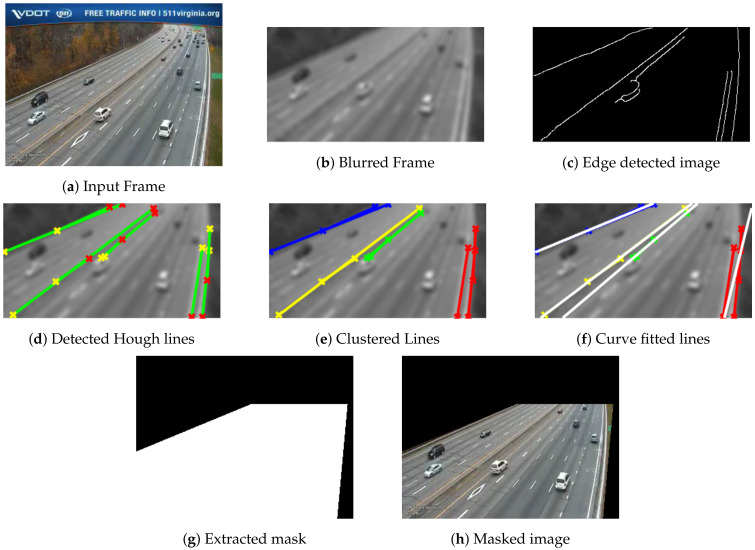
Different output for ROI extraction pipeline stages for Video 1.

**Figure 6 sensors-22-08121-f006:**
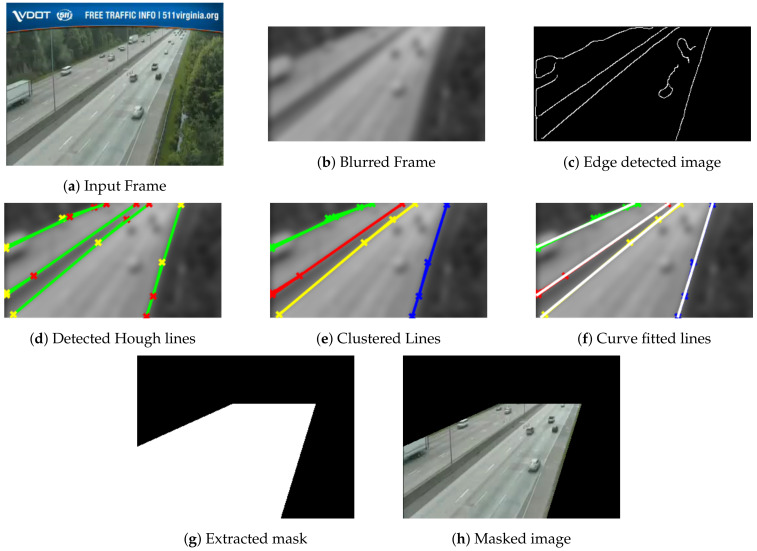
Different output for ROI extraction pipeline stages for Video 4.

**Figure 7 sensors-22-08121-f007:**
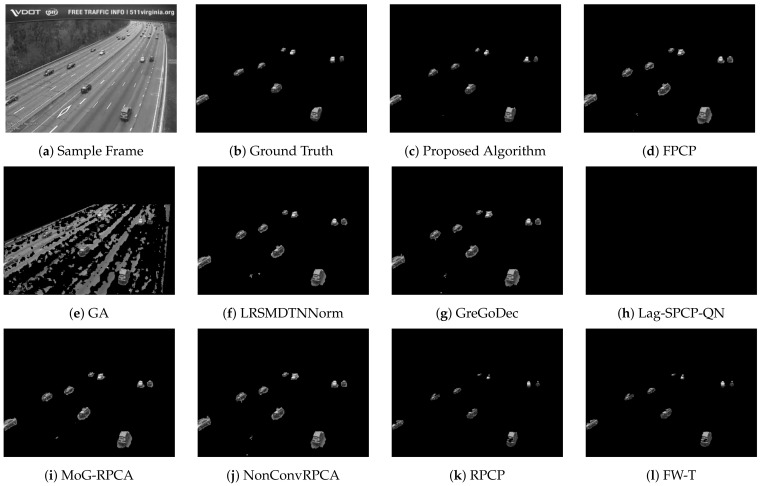
Video 1 segmentation outputs comparison for different segmentation techniques.

**Figure 8 sensors-22-08121-f008:**
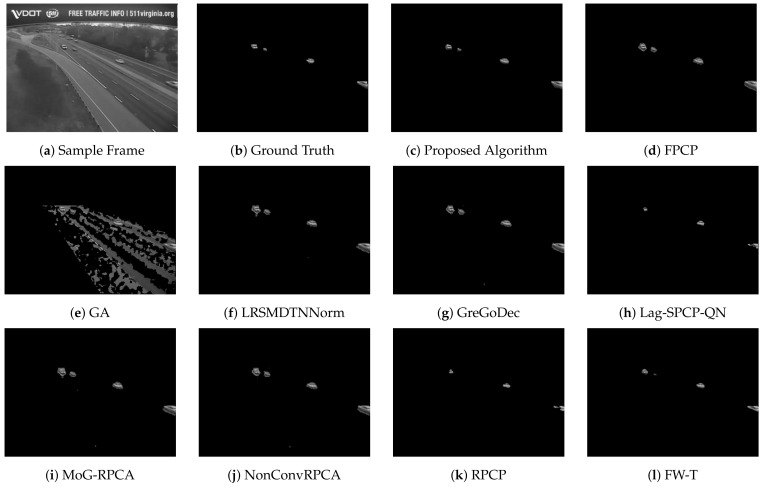
Video 2 segmentation outputs comparison for different segmentation techniques.

**Figure 9 sensors-22-08121-f009:**
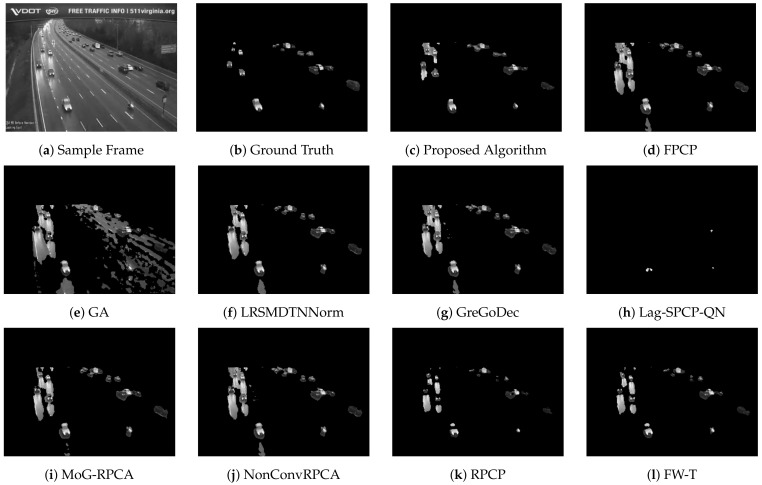
Video 3 segmentation outputs comparison for different segmentation techniques.

**Figure 10 sensors-22-08121-f010:**
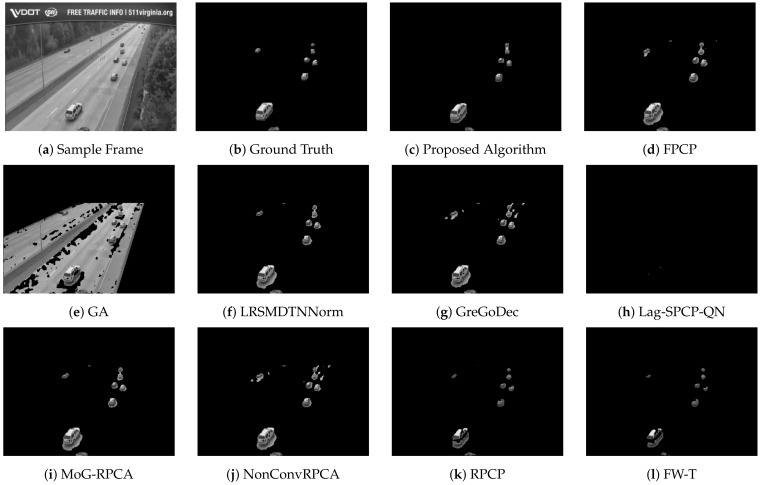
Video 4 segmentation outputs comparison for different segmentation techniques.

**Figure 11 sensors-22-08121-f011:**
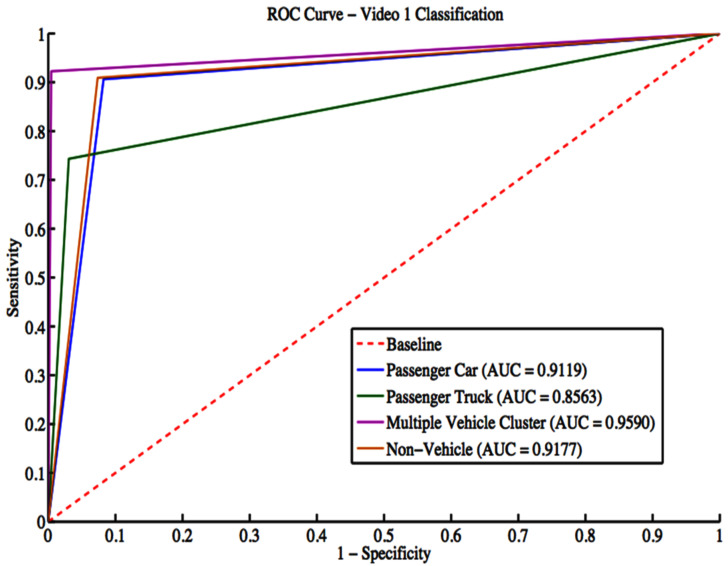
ROC curves for Video 1 classification.

**Figure 12 sensors-22-08121-f012:**
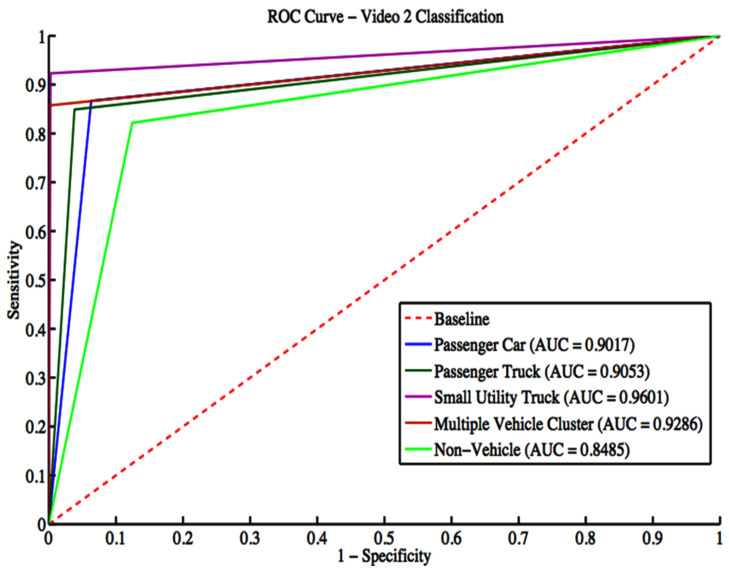
ROC curves for Video 2 classification.

**Figure 13 sensors-22-08121-f013:**
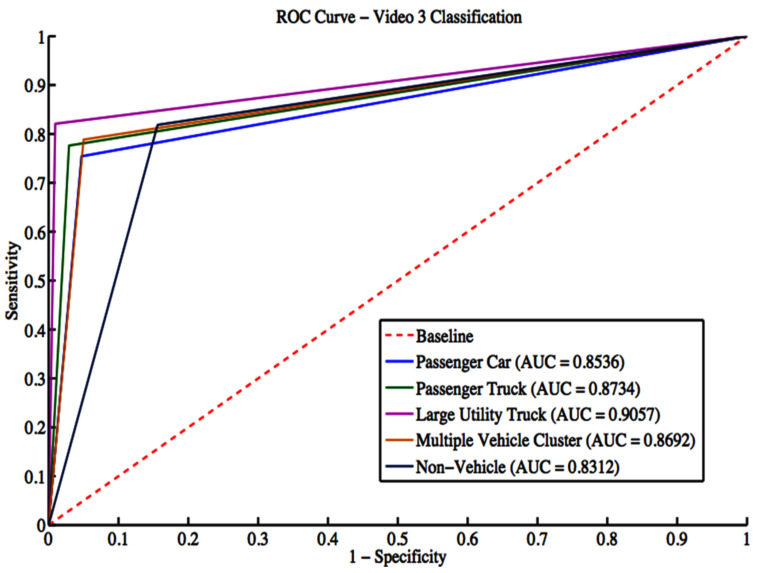
ROC curves for Video 3 classification.

**Figure 14 sensors-22-08121-f014:**
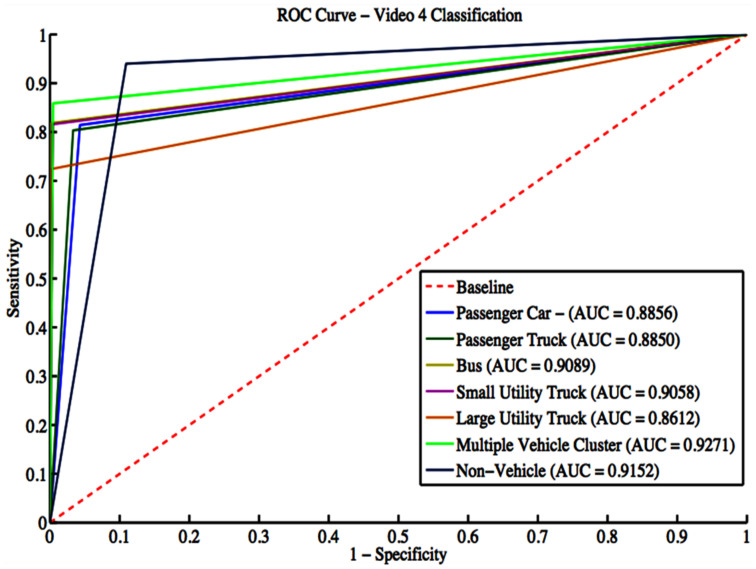
ROC curves for Video 4 classification.

**Table 1 sensors-22-08121-t001:** Description of sample videos for evaluation.

Name	Number of Frames	Time	Road Description	Environmental Description
Video 1	299	20 s	5 lanes in each direction, medium traffic	Sunny, daytime
Video 2	299	20 s	2 lanes in each direction and an oncoming onramp, light traffic	Low light, dusk
Video 3	299	20 s	5 lanes in each direction, heavy traffic	Rain, daytime
Video 4	899	60 s	4 lanes in each direction, heavy traffic	Sunny, daytime

**Table 2 sensors-22-08121-t002:** Proposed technique foreground segmentation evaluation against different recent algorithms for Video 1.

	Recall	CA	Precision	F-Score	Ave. Comp. Time (s)
Proposed Algorithm	0.8988	0.9909	0.8288	0.8619	0.0089
FPCP	0.9740	0.9820	0.6534	0.7815	0.0117
LRSMDTNNorm	0.9775	0.9810	0.6393	0.7724	0.579
GA	0.9852	0.5665	0.0681	0.1266	0.0111
GreGoDec	0.9766	0.9806	0.6357	0.7694	0.0239
RPCP	0.7374	0.9868	0.8322	0.7807	0.1288
FW-T	0.7788	0.9874	0.8226	0.7990	0.0487
MoG-RPCA	0.9783	0.9814	0.6457	0.7773	0.1659
Lag-SPCP-QN	0.0094	0.9684	0.9778	0.0215	0.4898
NonConvRPCA	0.9762	0.9805	0.6350	0.7688	0.0337

**Table 3 sensors-22-08121-t003:** Proposed technique foreground segmentation evaluation against different recent algorithms for Video 2.

	Recall	CA	Precision	F-Score	Ave. Comp. Time (s)
Proposed Algorithm	0.8413	0.9966	0.7728	0.8029	0.0022
FPCP	0.9504	0.9919	0.5277	0.6775	0.0119
LRSMDTNNorm	0.9637	0.9905	0.4827	0.6420	0.458
GA	0.9672	0.4425	0.0141	0.0277	0.0111
GreGoDec	0.9617	0.9904	0.4786	0.6376	0.0189
RPCP	0.3250	0.9943	0.9194	0.5137	0.1102
FW-T	0.5959	0.9953	0.8121	0.6733	0.0348
MoG-RPCA	0.9638	0.9901	0.4718	0.6324	0.1054
Lag-SPCP-QN	0.3401	0.9940	0.8834	0.4595	0.2327
NonConvRPCA	0.9617	0.9904	0.4786	0.6376	0.0263

**Table 4 sensors-22-08121-t004:** Proposed technique foreground segmentation evaluation against different recent algorithms for Video 3.

	Recall	CA	Precision	F-Score	Ave. Comp. Time (s)
Proposed Algorithm	0.7003	0.9702	0.6345	0.6648	0.0022
FPCP	0.8549	0.9372	0.3991	0.5422	0.0150
LRSMDTNNorm	0.9687	0.9864	0.6159	0.7518	0.49
GA	0.8630	0.7411	0.1372	0.2339	0.0135
GreGoDec	0.8643	0.9346	0.3887	0.5343	0.0203
RPCP	0.5116	0.9630	0.5839	0.5414	0.1587
FW-T	0.6388	0.9610	0.5475	0.5870	0.0513
MoG-RPCA	0.8616	0.9417	0.4240	0.5663	0.3055
Lag-SPCP-QN	0.0195	0.9581	0.8531	0.0438	0.4203
NonConvRPCA	0.8643	0.9346	0.3887	0.5343	0.0241

**Table 5 sensors-22-08121-t005:** Proposed technique foreground segmentation evaluation against different recent algorithms for Video 4.

	Recall	CA	Precision	F-Score	Ave. Comp. Time (s)
Proposed Algorithm	0.8751	0.9927	0.8202	0.8440	0.0067
FPCP	0.9621	0.9876	0.6370	0.7652	0.0112
LRSMDTNNorm	0.9687	0.9864	0.6159	0.7518	0.626
GA	0.9620	0.1557	0.0254	0.0491	0.0104
GreGoDec	0.9670	0.9862	0.6064	0.7436	0.0211
RPCP	0.7130	0.9897	0.8093	0.7551	0.1468
FW-T	0.5956	0.9885	0.8526	0.6979	0.0381
MoG-RPCA	0.9397	0.9847	0.6104	0.7368	0.1865
Lag-SPCP-QN	0.0543	0.9773	0.9554	0.0964	0.8091
NonConvRPCA	0.9670	0.9862	0.6064	0.7436	0.0441

**Table 6 sensors-22-08121-t006:** Confusion matrix for Video 1 for all frames in ROI.

		Predicted Class			
		Passenger Car	Passenger Truck	Multiple Vehicle Cluster	Non-Vehicle
**Actual Class**	Passenger Car	474	18	0	31
	Passenger Truck	29	191	4	33
	Multiple Vehicle Cluster	5	2	119	3
	Non-Vehicle	6	3	0	90

**Table 7 sensors-22-08121-t007:** Different evaluation metrics for the detected classes within Video 1.

Vehicle Class	Precision	Recall	F-Score
Passenger Car	92%	91%	91%
Passenger Truck	89%	74%	81%
Multiple Vehicle Cluster	97%	92%	94%
Non-Vehicle	57%	91%	70%

**Table 8 sensors-22-08121-t008:** Multiple vehicle cluster detections confusion matrix for Video 1.

		Predicted Class		
		Passenger Car	Passenger Truck	Non-Vehicle
Actual Class	Passenger Car	120	0	24
	Passenger Truck	24	72	24
	Non-Vehicle	0	0	0

**Table 9 sensors-22-08121-t009:** Different evaluation metrics for detected vehicles in multiple vehicle clusters within Video 1.

Vehicle Class	Precision	Recall	F-Score
Passenger Car	83%	83%	83%
Passenger Truck	99%	60%	75%
Non-Vehicle	0	0	0

**Table 10 sensors-22-08121-t010:** Confusion matrix for all vehicles in Video 2.

		Predicted Class				
		Passenger Car	Passenger Truck	Small Utility Truck	Multiple Vehicle Cluster	Non-Vehicle
Actual Class	Passenger Car	104	1	0	0	15
	Passenger Truck	0	73	1	0	12
	Small Utility Truck	0	1	12	0	0
	Multiple Vehicle Cluster	0	0	0	12	2
	Non-Vehicle	16	9	0	0	115

**Table 11 sensors-22-08121-t011:** Different evaluation metrics for the detected classes within Video 2.

Vehicle Class	Precision	Recall	F-score
Passenger Car	87%	87%	87%
Passenger Truck	87%	85%	86%
Small Utility Truck	92%	92%	92%
Multiple Vehicle Cluster	99%	86%	92%
Non-Vehicle	80%	82%	81%

**Table 12 sensors-22-08121-t012:** Multiple vehicle cluster detections confusion matrix for Video 2.

		Predicted Class			
		Passenger Car	Passenger Truck	Small Utility Truck	Non-Vehicle
Actual Class	Passenger Car	13	0	1	1
	Passenger Truck	1	5	1	2
	Small Utility Truck	2	0	0	1
	Non-Vehicle	0	0	0	0

**Table 13 sensors-22-08121-t013:** Different evaluation metrics for detected vehicles in multiple vehicle clusters within Video 2.

Vehicle Class	Precision	Recall	F-Score
Passenger Car	81%	87%	84%
Passenger Truck	99%	56%	71%
Small Utility Truck	0	0	0
Non-Vehicle	0	0	0

**Table 14 sensors-22-08121-t014:** Confusion matrix for all vehicles in Video 3.

		Predicted Class							
		Passenger Car	Passenger Truck	Motorcycle	Bus	Small Utility Truck	Large Utility Truck	Multiple Vehicle Cluster	Non-Vehicle
Actual Class	Passenger Car	350	13	0	0	0	0	9	92
	Passenger Truck	16	371	0	0	0	0	13	78
	Motorcycle	0	0	0	0	0	0	1	0
	Bus	0	0	0	0	0	0	0	2
	Small Utility Truck	0	0	0	1	0	1	0	1
	Large Utility Truck	0	0	0	0	6	55	1	5
	Multiple Vehicle Cluster	5	9	0	0	0	16	149	10
	Non-Vehicle	49	21	0	0	0	1	64	609

**Table 15 sensors-22-08121-t015:** Different evaluation metrics for the detected classes within Video 3.

Vehicle Class	Precision	Recall	F-Score
Passenger Car	83%	75%	79%
Passenger Truck	90%	78%	83%
Motorcycle	0	0	0
Bus	0	0	0
Small Utility Truck	0	0	0
Large Utility Truck	75%	82%	79%
Multiple Vehicle Cluster	63%	79%	70%
Non-Vehicle	76%	82%	79%

**Table 16 sensors-22-08121-t016:** Confusion matrix for multiple vehicle cluster detections for Video 3.

		Predicted Class						
		Passenger Car	Passenger Truck	Motorcycle	Bus	Small Utility Truck	Large Utility Truck	Non-Vehicle
Actual Class	Passenger Car	263	2	0	0	2	5	55
	Passenger Truck	1	72	1	2	1	2	15
	Motorcycle	0	0	0	0	0	0	21
	Bus	4	2	5	0	0	2	23
	Small Utility Truck	0	11	4	1	0	12	31
	Large Utility Truck	3	0	0	0	1	1	6
	Non-Vehicle	0	0	0	0	0	0	0

**Table 17 sensors-22-08121-t017:** Different evaluation metrics for detected vehicles in multiple vehicle clusters within Video 3.

Vehicle Class	Precision	Recall	F-Score
Passenger Car	97%	80%	88%
Passenger Truck	83%	77%	80%
Motorcycle	0	0	0
Bus	0	0	0
Small Utility Truck	0	0	0
Large Utility Truck	4.5%	9.1%	6.1%
Non-Vehicle	0	0	0

**Table 18 sensors-22-08121-t018:** Confusion matrix for all vehicles in Video 4.

		Predicted Class						
		Passenger Car	Passenger Truck	Bus	Small Utility Truck	Large Utility Truck	Multiple Vehicle Cluster	Non-Vehicle
Actual Class	Passenger Car	482	30	0	0	0	3	77
	Passenger Truck	61	539	1	1	0	7	62
	Bus	0	1	18	0	0	0	3
	Small Utility Truck	2	5	0	102	0	1	15
	Large Utility Truck	0	2	0	5	42	0	9
	Multiple VehicleCluster	5	7	0	2	0	231	24
	Non-Vehicle	30	28	0	4	5	1	1063

**Table 19 sensors-22-08121-t019:** Different evaluation metrics for the detected classes within Video 4.

Vehicle Class	Precision	Recall	F-score
Passenger Car	83%	81%	82%
Passenger Truck	88%	80%	84%
Bus	95%	82%	88%
Small Utility Truck	89%	82%	85%
Large Utility Truck	89%	72%	80%
Multiple Vehicle Cluster	95%	86%	90%
Non-Vehicle	85%	94%	89%

**Table 20 sensors-22-08121-t020:** Confusion matrix for multiple vehicle cluster detections for Video 4.

		Predicted Class					
		Passenger Car	Passenger Truck	Bus	Small Utility Truck	Large Utility Truck	Non-Vehicle
Actual Class	Passenger Car	216	7	0	13	9	3
	Passenger Truck	13	108	0	5	11	1
	Bus	2	9	2	0	3	6
	Small Utility Truck	3	7	2	41	5	1
	Large Utility Truck	31	2	0	0	7	13
	Non-Vehicle	0	0	0	0	0	0

**Table 21 sensors-22-08121-t021:** Different evaluation metrics for detected vehicles in multiple vehicle clusters within Video 4.

Vehicle Class	Precision	Recall	F-score
Passenger Car	82%	87%	84%
Passenger Truck	81%	78%	80%
Bus	50%	9.1%	15%
Small Utility Truck	69%	69%	69%
Large Utility Truck	20%	13%	16%
Non-Vehicle	0	0	0

**Table 22 sensors-22-08121-t022:** Different evaluated DL architectures with different feature extractor networks.

	Faster R-CNN	YOLO 2	YOLO 3	YOLO 4 COCO	YOLO 4 Tiny COCO
Resnet 18	*√*	*√*	*√*		
Resnet 50	*√*	*√*	*√*		
Restnet 101	*√*	*√*	*√*		
Inception V3	*√*	*√*	*√*		
Inception resnet V2	*√*	*√*	*√*		
Darknet 53		*√*	*√*	*√*	*√*
Squeeznet		*√*	*√*		
Goolgenet		*√*	*√*		

**Table 23 sensors-22-08121-t023:** Average precision for tested techniques.

Tested Algorithm	Video 1	Video 2	Video 3	Video 4
YOLO V2—resnet18	0.42	0.38	0.35	0.40
YOLO V2—resnet50	0.47	0.39	0.40	0.40
YOLO V2—resnet101	0.50	0.43	0.42	0.45
YOLO V2—inceptionv3	0.51	0.44	0.39	0.43
YOLO V2—inceptionresnetv2	0.52	0.43	0.40	0.47
YOLO V2—darknet19	0.57	0.46	0.36	0.55
YOLO V2—squeeznet	0.44	0.43	0.40	0.42
YOLO V2—googlenet	0.44	0.39	0.33	0.41
YOLO V2—darknet53	0.56	0.52	0.41	0.53
YOLO V3—resnet18	0.52	0.52	0.35	0.50
YOLO V3—resnet50	0.53	0.41	0.39	0.47
YOLO V3—resnet101	0.58	0.43	0.46	0.56
YOLO V3—inceptionv3	0.47	0.46	0.39	0.44
YOLO V3—inceptionresnetv2	0.41	0.40	0.35	0.45
YOLO V3—squeeznet	0.55	0.53	0.42	0.52
YOLO V3—googlenet	0.53	0.50	0.43	0.46
YOLO V3—darknet53	0.58	0.54	0.44	0.56
YOLO V4 COCO—darknet53	0.61	0.58	0.46	0.59
YOLO V4 Tiny COCO—darknet53	0.51	0.50	0.41	0.52
Faster RCNN—resnet18	0.45	0.42	0.35	0.44
Faster RCNN—resnet50	0.45	0.40	0.31	0.44
Faster RCNN—resnet101	0.49	0.42	0.35	0.48
Faster RCNN—darknet53	0.51	0.48	0.38	0.47
Proposeds Algorithm	0.85	0.83	0.71	0.81

**Table 24 sensors-22-08121-t024:** Summary of timing analysis for each sample video.

Name	Number of Frames	Run Time (s)	Average Frame Rate (s)
Video 1	299	22.724	0.076
Video 2	299	20.123	0.0673
Video 3	299	16.475	0.0551
Video 4	599	55.108	0.092

## Data Availability

Not applicable.

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
