# Peer review of "Monocular Camera Viewpoint-Invariant Vehicular Traffic Segmentation and Classification Utilizing Small Datasets"

_sensors, 2022, doi:10.3390/s22218121_

Round 1

Reviewer 1 Report

This paper shows the vision-based framework that is capable of segmenting moving vehicles and classifying.The proposed method has good performance and usefulness compared with other method. This paper is well organized and seems good for publishing.

1.Figure 3, 5,6,7,8,9,10. Characters are not displayed correctly.

2.The explanation of Table10 is unclear.

Reviewer 2 Report

In the article “Monocular camera viewpoint-invariant vehicular traffic segmentation and classification utilizing small datasets”, the authors developed and implemented a monocular camera viewpoint independent and integrated method for vehicular traffic segmentation and classification. They evaluated their results using networked traffic camera videos obtained from the Virginia Department of Transportation.

Given the expediency of providing a report, my comments are somewhat limited, though I hope they are still useful to the editors and authors:

The research topic  is of importance and worthy of investigation. Overall the article is good, the methodology and procedure appear sound and the results are interesting. This paper can be accepted after necessary revisions. The following issues must be addressed and clarified before acceptance of the article.

1.      The novelty of the work must be clearly mentioned in the abstract of the article as well as in the conclusion section.

2.      I think there is no need for a separate literature review section, it is suggested to merged its contents in the introduction section instead.

3.      The methods part is well explained. Normally, the procedure/methodology must be comprehensive and detailed enough to be reproduced by other researchers which the authors have presented very well and their efforts are appreciated. However, if the authors think that there are some details missing which might be useful to young researchers to reproduce the results can be further added.

4.      In the classification model, the accuracy could also be high due to class imbalance in the dataset. Could the authors provide more information on the distribution of classes by providing information about confusion matrix? In case there is significant class imbalance, it might also aid their analysis if they perform under/over sampling or selectively penalize misclassification of the minority class.

5.      It will further enhance the quality of this work if the current results are compared with some more latest relevant studies.

6.      I noticed several grammatical/ sentence structuring mistakes throughout the paper.  It is suggested to thoroughly check the manuscript and correct such type of errors. Overall the English write up is somewhat weak and need to be improved further.

Reviewer 3 Report

The paper is suitable for publication. The following points have to be corrected.

1) You report high accurate results using small data set. However, you have to explain why your proposed algorithm can improve the accuracy using small data set in section 3.

2) Describe how you reduce the data set size in the result. If possible, please add the data set size in Tables 2-5.
